Computational Biology

METHODS

# Semi-parametric empirical bayes method for multiplet detection in snATAC-seq with probabilistic multi-omic integration

Yuntian Wu[1]*, Haoran Hu[2], Wei Chen[2,3,4], Johann E. Gudjonsson[5,6], Lam C. Tsoi[1,5,7,8]*, Xiaoquan Wen[1]*

1 Department of Biostatistics, University of Michigan, Ann Arbor, Michigan, United States of America, 2 Department of Biostatistics, University of Pittsburgh, Pittsburgh, Pennsylvania, United States of America, 3 Department of Pediatrics, University of Pittsburgh, Pittsburgh, Pennsylvania, United States of America, 4 Department of Human Genetics, University of Pittsburgh, Pittsburgh, Pennsylvania, United States of America, 5 Department of Dermatology, University of Michigan, Ann Arbor, Michigan, United States of America, 6 Department of Internal Medicine, Division of Rheumatology, University of Michigan, Ann Arbor, Michigan, United States of America, 7 Mary H. Weiser Food Allergy Center, University of Michigan, Ann Arbor, Michigan, United States of America, 8 Department of Computational Medicine and Bioinformatics, University of Michigan, Ann Arbor, Michigan, United States of America

* yuntian@umich.edu (YW); alextsoi@med.umich.edu (LCT); xwen@umich.edu (XW)

## Abstract

Multiplets arise when multiple cells are captured within the same droplet during single-cell sequencing, producing hybrid molecular profiles that can distort downstream analyses. Detecting multiplets in single-nucleus ATAC-seq (snATAC-seq) data is particularly challenging due to the sparsity and overdispersion of chromatin accessibility measurements. Moreover, computational approaches that jointly leverage evidence across multiple features and data modalities are highly desirable for multiplet detection. We introduce SEBULA, a semi-parametric empirical Bayes framework for multiplet detection in snATAC-seq data. SEBULA models the singlet background directly from observed chromatin accessibility signals using fragment-level information from snATAC-seq data. This approach avoids reliance on synthetic doublets and produces classification probabilities that enable direct false discovery rate control. We further extend SEBULA to integrate complementary evidence from additional features and modalities, such as simultaneously measured gene expression profiles. Across simulations and seven multimodal datasets with hashing-based ground truth, SEBULA demonstrates improved sensitivity and specificity compared with existing snATAC-seq methods. The evidence integration framework achieves comparable or superior performance relative to state-of-the-art multiomic approaches while maintaining computational efficiency.

**Data availability statement:** All datasets analyzed in this study were previously published and are publicly available. The DOGMA-seq dataset is available from the Gene Expression Omnibus (GEO) under accession number GSE200417. The six PBMC datasets ("PB-1", "PB-2", "PB-3", "PB-4", "PB-8", "PB-9") were generated and published in the COMPOSITE study (Hu et al., 2024), and are publicly available at: https://zenodo.org/records/11167173. The SEBULA software is freely available at https://github.com/Yuntian0716/SEBULA. The repository also provides the scripts and instructions used for the simulations and data processing in this study.

**Funding:** This work is supported by National Institutes of Health (R35GM138121 to XW; R01ES033634 to XW; R01AR080662 to LCT; 1P30AR075043 to LCT. and JEG; UC2 AR081033 to LCT and JEG.). The funders had no role in study design, data collection and analysis, decision to publish, or preparation of the manuscript.

**Competing interests:** I have read the journal's policy and the authors of this manuscript have the following competing interests: JEG has served as a consultant to AbbVie, Eli Lilly, Almirall, Celgene, BMS, Janssen, Prometheus, TimberPharma, Galderma, Novatis, MiRagen, AnaptysBio, and has received research support from AbbVie, SunPharma, Eli Lilly, Kyowa Kirin, Almirall, Celgene, BMS, Janssen, Prometheus, and TimberPharma. LCT has received support from Galderma and Janssen.

## Author summary

Single-cell sequencing has revolutionized biology by allowing researchers to look at the genetic activity of thousands of individual cells simultaneously. However, common technical artifacts occur when two or more cells are accidentally trapped in the same reaction droplet. These "multiplets" create a blurred, hybrid signal that can lead researchers to false biological conclusions. Detecting these artifacts is especially difficult in data that measures chromatin accessibility (i.e., the openness of DNA), which is often sparse and noisy. We developed SEBULA, a new computational tool designed to solve this problem. Unlike existing methods that rely on simulated data to guess what a multiplet looks like, SEBULA learns the characteristics of true single cells directly from the observed data. This makes it more accurate at spotting subtle multiplet signals that other tools miss. Furthermore, SEBULA is built for the latest multimodal technologies that measure different types of biological information at once. It can combine evidence from multiple sources, such as gene activity and DNA structure, to confirm if a droplet contains a single cell or multiple cells. By providing a more reliable way to identify and remove multiplets, SEBULA helps improve the reliability of downstream analyses in single-cell studies.

## Introduction

Single-cell sequencing technologies have transformed our ability to characterize cellular heterogeneity by measuring molecular features at the single-cell level. However, a persistent technical challenge is the formation of multiplets, in which two or more cells or nuclei are co-encapsulated within a single droplet and subsequently processed as a single unit [1–3]. Because multiplets generate hybrid transcriptomic and epigenomic profiles that do not correspond to genuine biological states, they can produce spurious signals in many downstream analyses, including clustering, differential expression, trajectory inference, and allele-specific accessibility [3–5]. Multiplets can be classified as heterotypic or homotypic, depending on whether the encapsulated cells originate from different or the same (or highly similar) transcriptional states. While heterotypic multiplets are well known to introduce severe artifacts in downstream analyses, homotypic doublets are sometimes regarded as relatively innocuous in certain standard single-cell workflows [4]. Nevertheless, they can still adversely affect specific downstream analyses. For example, filtering homotypic multiplets is critical for studies involving allelic bias and lineage tracing, where aggregated fragment counts can distort per-cell allele-specific or lineage signals [5]. In addition, both types of multiplets may alter transcriptional dropout patterns and thereby influence transcriptomic imputation methods [6]. As single-cell experiments continue to scale in both throughput and complexity, accurate detection and removal of multiplets have become critical upstream quality-control steps in single-cell analysis pipelines, helping ensure the reliability of downstream biological conclusions.

Complementary experimental and genotype-based approaches, such as Cell Hashing [7], MULTI-seq [8], and demuxlet [9], can identify inter-sample multiplets and provide orthogonal benchmarks for doublet detection. However, these methods require additional experimental steps and costs, cannot be applied retrospectively to unlabeled datasets, and generally fail to detect within-sample (same-donor) multiplets. As a result, computational strategies that operate directly on single-cell molecular profiles remain essential for comprehensive multiplet detection.

The prevailing paradigm for computational multiplet detection is based on artificial doublet simulation. Approaches such as Scrublet [10], scds [11], DoubletFinder [6], Solo [12], and scDblFinder [13] generate artificial doublets *in silico* by summing or averaging molecular profiles, and then identify real cells that localize near these simulated profiles in low-dimensional spaces. This framework has been highly influential and performs well in transcriptomic data, particularly for detecting heterotypic multiplets [6,10]. However, the success of simulation-based approaches relies heavily on the assumption that artificial mixtures adequately approximate the distribution of true multiplets. In practice, real multiplets may deviate from simple additive behavior due to technical and biological factors [14]. More importantly, homotypic multiplets are particularly difficult to detect from RNA profiles alone [6]. To address this limitation, approaches such as ArchR [15] and scDblFinder [13] extend simulation-based strategies to single-nucleus ATAC-seq (snATAC-seq). Nevertheless, snATAC-seq data present additional challenges for this simulation-based paradigm: chromatin accessibility data are sparse, often reflect discrete accessibility states (e.g., 0 for closed chromatin, 1 for open on one parental chromosome, and 2 for open on both), and exhibit scattered, low-coverage signals. In contrast, AMULET [5] introduces a model-based approach that leverages fragment-level information and shows good performance in detecting both heterotypic and homotypic doublets. However, AMULET imposes rather strong distributional assumptions by assuming a homogeneous singlet population, whereas practical snATAC-seq datasets often exhibit heterogeneous sequencing depth, variable sample composition, and overdispersion.

Recently, multiomic platforms such as DOGMA-seq [16], CITE-seq [17], TEA-seq [18], SNARE-seq [19], and SHARE-seq [20] have enabled the simultaneous measurement of gene expression, chromatin accessibility, and protein abundance across thousands of individual cells [21]. These data provide complementary evidence for multiplet detection: a multiplet may appear ambiguous in one modality but display a strong signal in another. This motivates integration strategies that combine modality-specific evidence. Computational methods such as COMPOSITE [14] and OmniDoublet [22] have shown great promise for jointly modeling evidence across multiple data modalities to detect multiplets. Nevertheless, alternative inference strategies that directly incorporate existing modality-specific evidence remain attractive, as they are computationally efficient and can accommodate diverse data types and experimental designs.

In this paper, we present SEBULA (**S**emi-parametric **E**mpirical **B**ayes approach for m**UL**tiplet detection in single-nucleus **A**TAC-seq), a read-based, ATAC-focused computational framework for multiplet detection. SEBULA uses fragment-level high-coverage loci count (HCLC) statistics derived from snATAC-seq data [5] and learns the distributions of singlets and multiplets by employing a semi-parametric empirical Bayes framework, thereby avoiding reliance on artificial doublet simulation or strong parametric assumptions. Building on this ATAC-based core, we further introduce an integration strategy that combines SEBULA's chromatin-based evidence with complementary signals from other features or data modalities within an established machine learning framework. SEBULA is freely available as an open-source software package at https://github.com/Yuntian0716/SEBULA.

## Results

### Overview

SEBULA implements a semi-parametric probabilistic model for multiplet detection in snATAC-seq data (Fig 1). Following a preprocessing strategy similar to AMULET, we generate a binary matrix of multi-read sites, where rows correspond to genomic loci with more than two overlapping reads in at least one cell (i.e., high-coverage loci), and columns represent

PLOS Computational Biology

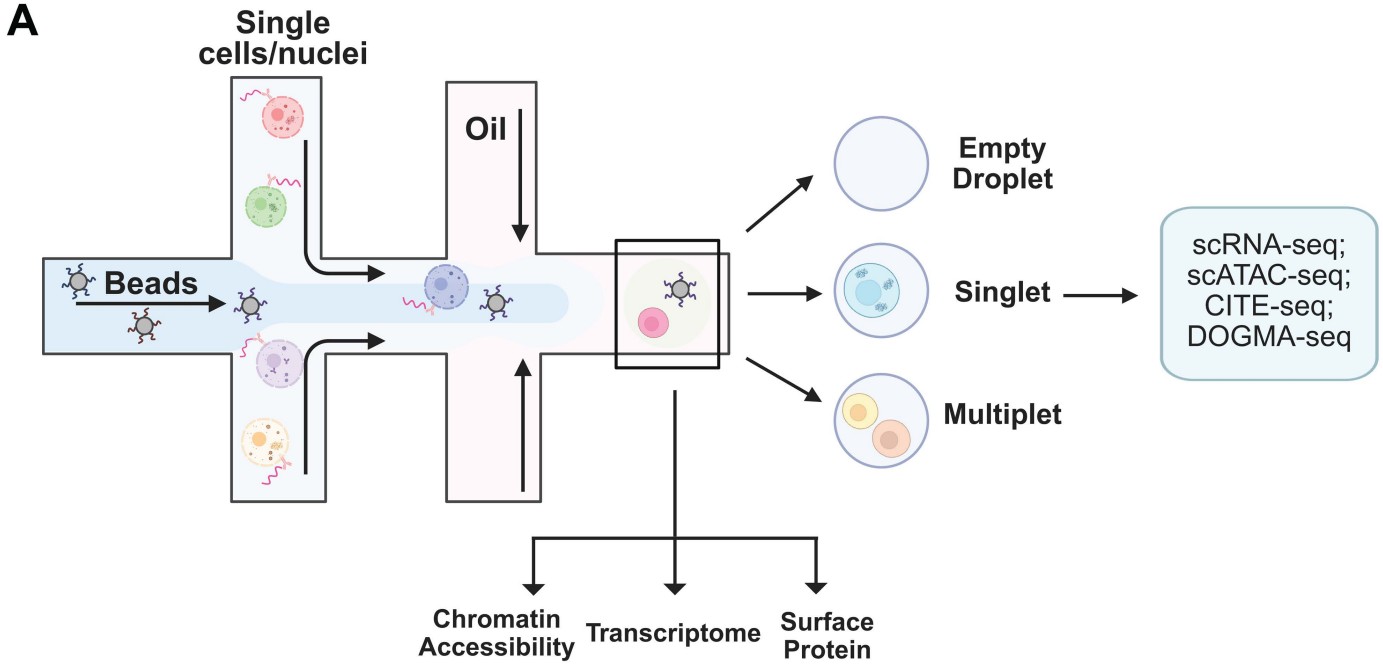

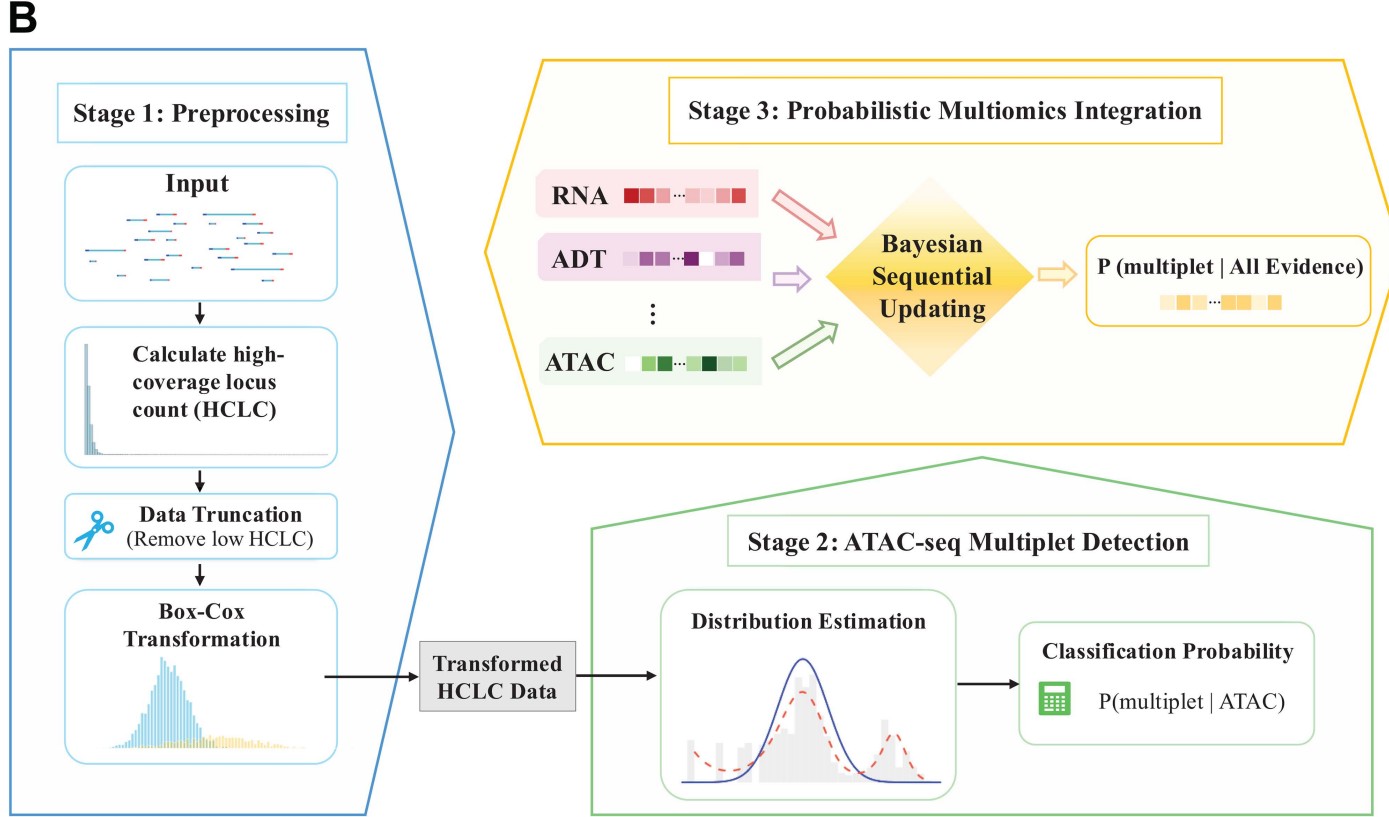

**Fig 1. Overview of the SEBULA framework. (A) Formation of multiplets.** In droplet-based platforms, individual cells or nuclei are encapsulated together with barcoded beads in oil droplets. Ideally, each droplet contains a single cell (singlet), but some droplets capture multiple cells (multiplets) or remain empty. Multiplets generate hybrid molecular profiles that can confound downstream analyses. Modern single-cell assays, including single-cell

RNA sequencing, single-nucleus ATAC sequencing, CITE-seq, and multimodal platforms such as DOGMA-seq, measure complementary molecular layers, including the transcriptome, chromatin accessibility, and surface protein abundance (Created in BioRender. Tsoi, A. L. (2026) https://BioRender. com/xrpr281). **(B) Computational workflow.** SEBULA operates in three stages. In Stage 1 (preprocessing), fragment-level chromatin accessibility data are used to compute the high-coverage locus count (HCLC). Cells with very low HCLC values are truncated to stabilize estimation, and a Box-Cox transformation is applied to the remaining HCLC data. In Stage 2, the transformed HCLC distribution is modeled using a nonparametric empirical Bayes framework to estimate the singlet distribution and derive a posterior probability of multiplet status based on ATAC-seq evidence. In Stage 3, when evidence from additional data features and modalities is available, SEBULA integrates feature-specific probabilistic evidence and yields a combined posterior probability of multiplet status given all available measurements.

individual nuclei. From this matrix, we compute the high-coverage locus count (HCLC) for each cell, which serves as input to SEBULA. Intuitively, cells with unusually high HCLC values are more likely to be multiplets.

The key innovation of SEBULA lies in its semi-parametric procedure for directly estimating the distributions of HCLC in singlets and multiplets from observed data. Real snATAC-seq datasets with ground-truth multiplet annotations suggest that commonly used parametric families for count data, such as the Poisson (used by AMULET) and negative binomial distributions, fit the empirical HCLC data poorly (Fig 2). Informed by these empirical observations, we reason that effective multiplet detection should prioritize distinguishing multiplets from singlets with moderate to high HCLC values, since cells with extremely low HCLC values are most likely singlets and therefore provide limited information for multiplet detection. To achieve this, we adopt a strategy analogous to the *zero assumption* in non-parametric empirical Bayes (NPEB) inference [23], focusing on accurate characterization of the right-tail behavior of the singlet distribution. Specifically, cells with very low HCLC values (defined by a user-specified threshold) are excluded, and the remaining counts are transformed to a continuous domain using a Box-Cox transformation. The empirical density of the transformed values, denoted by $f(x)$, is then estimated and modeled as a mixture of singlet and multiplet distributions:

$$f(x) = \pi_0 f_0(x) + \pi_1 f_1(x),$$

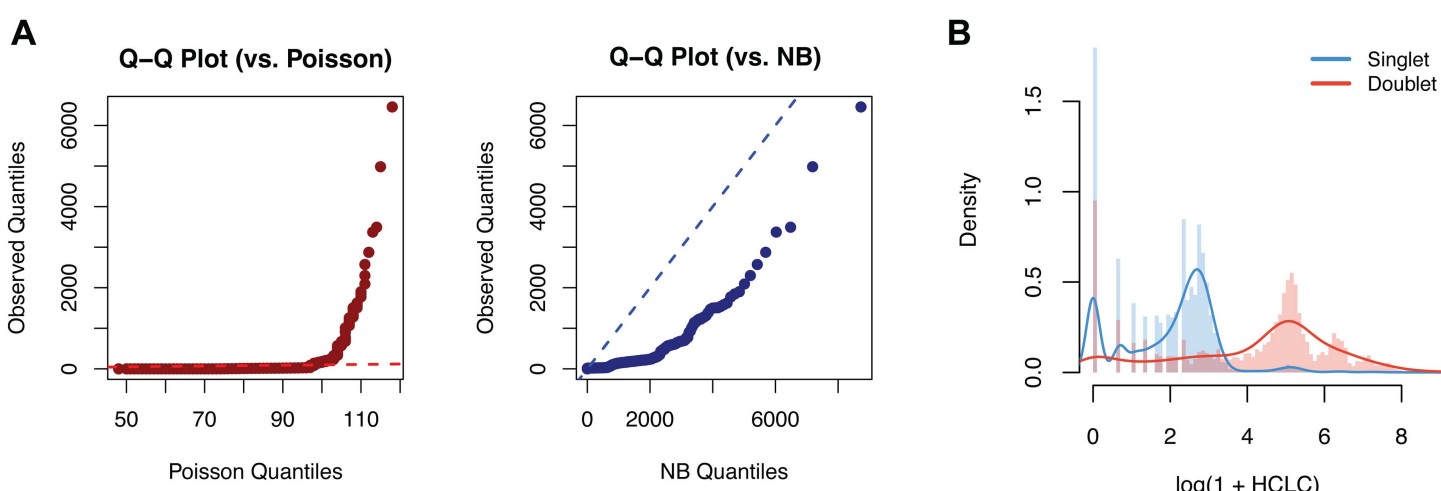

**Fig 2. Commonly-used parametric models fit poorly to singlet HCLC data.** Plotting data are from a DOGMA-seq dataset, with ground-truth singlet and multiplet labels obtained via cell hashing. **(A)** Quantile–quantile (Q–Q) plots comparing the observed singlet distribution with the quantiles of fitted values under Poisson and negative binomial models. Neither parametric model adequately fits the data. **(B)** Histograms of log(1 + HCLC) values for singlet (light blue) and doublet (red) nuclei in a DOGMA-seq dataset (PB-3), stratified by ground-truth labels. Kernel density estimates are overlaid. The figure shows that singlets and doublets become separable after transformation; however, no simple parametric model adequately captures the empirical distributions.

where $x$ denotes the transformed count, and $\pi_0$ and $\pi_1$ represent the proportions of singlets and multiplets, respectively.

Following the principle of the zero assumption, we make two key assumptions: (i) singlets are substantially more abundant than multiplets ($\pi_0 \gg \pi_1$), and (ii) cells near the mode of the empirical distribution are predominantly singlets. Leveraging these assumptions, we apply non-parametric density estimation around the mode of $f(x)$ to infer the singlet distribution $f_0(x)$ and estimate its mixing proportion $\pi_0$ (Methods). The multiplet distribution $f_1(x)$ and its mixing proportion $\pi_1$ are subsequently induced. With these distributional estimates, the posterior probability that a cell is a multiplet is computed via Bayes' rule:

$$P(\text{multiplet} \mid x) = 1 - P(\text{singlet} \mid x) = 1 - \frac{\pi_0 f_0(x)}{f(x)}.$$

(1)

These estimates also enable a principled framework for local and Bayesian FDR control procedures in multiplet detection. Specifically, under a decision rule that classifies all cells with HCLC $\geq t$ as multiplets, the corresponding estimated false discovery rate (FDR) is given by

$$\text{FDR}(t) = P(\text{singlet} \mid x \geq t) = \frac{\pi_0 P(x \geq t \mid \text{singlet})}{P(x \geq t)}.$$

(2)

The detailed procedures for FDR and lfdr computation are provided in Section 2 of S1 Text.

Quantifying multiplet evidence from the HCLC on a probabilistic scale allows seamless integration with other probabilistic signals, whether derived from additional features in snATAC-seq data, or from complementary modalities such as scRNA-seq. Consequently, SEBULA can be combined with other multiplet detection tools that generate probabilistic outputs, including scDblFinder and COMPOSITE, to further enhance sensitivity and specificity.

This integration follows the principle of Bayesian sequential updating [24]. Specifically, the Bayes factor for the HCLC data, BF($x$), can be obtained directly from the posterior probability (1) given the snATAC-seq HCLC data $x$. By treating the posterior probability derived from additional feature data $y$, $P(\text{multiplet} \mid y)$, as a feature-informed prior, we construct a naïve Bayes classifier that integrates evidence across both sources. This yields the updated classification posterior:

$$P(\text{multiplet} \mid y, x) \propto P(\text{multiplet} \mid y) \cdot \text{BF}(x).$$

(3)

The integration step operates on modality-specific scores and does not require joint modeling of high-dimensional feature spaces. Additionally, this procedure above can be modified to integrate evidence in the form of $p$-values or $z$-scores. The details are provided in the Methods section and Section 3 of S1 Text.

## Evaluation using simulated artificial doublets

We generated synthetic snATAC-seq data to examine the performance of SEBULA and AMULET using only HCLC-derived information. Starting from a real DOGMA-seq dataset from Xu *et al.* [16], we extracted the scATAC-seq modality and removed all annotated multiplets. Artificial doublets were then created by randomly pairing the remaining 15,788 singlet cells. To assess performance across a range of practical scenarios, we constructed multiple datasets with artificial doublet proportions ranging from 5% to 25%. For each simulated dataset, the processed HCLC values were provided as input to both SEBULA and AMULET.

We evaluated the sensitivity and specificity of each method using receiver operating characteristic (ROC) curves and F1 scores across all simulated datasets. SEBULA consistently outperformed AMULET by a wide margin under all conditions (Figs 3 and S1). Notably, the performance of AMULET declined as the proportion of doublets increased, likely due to its reliance on a global mean estimate to approximate the singlet distribution. In contrast, SEBULA exhibited robust performance across varying doublet proportions, reflected by consistently higher AUC values (S1 Fig).

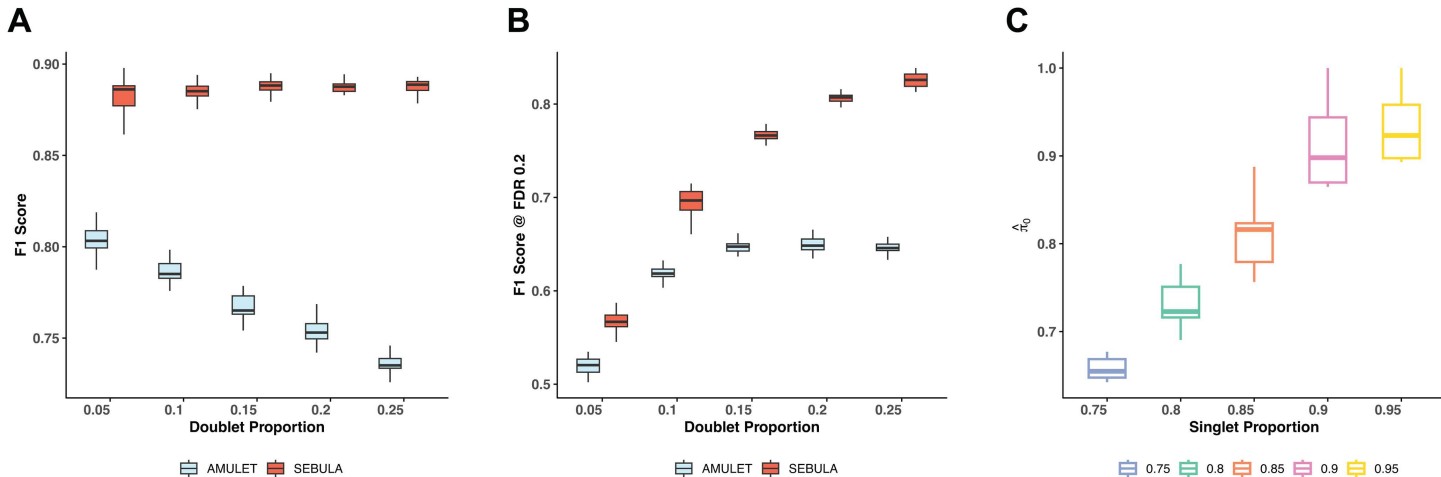

**Fig 3. Performance evaluation of SEBULA and AMULET using simulated doublet datasets.** In all panels, boxplots are stratified by the proportion of doublets in simulated datasets (5%–25%). Each boxplot summarizes results from 20 replicate simulations per condition. **(A)** Optimal F1 scores. **(B)** F1 scores evaluated at an FDR control level of 0.2. **(C)** Estimated singlet proportions ($\hat{\pi}_0$).

Both SEBULA and AMULET can estimate the false discovery rate (FDR) to determine a multiplet classification threshold, using Eqn. (2) and the built-in $q$-value method, respectively. At the 20% FDR control level, Fig 3B shows that SEBULA achieves substantially better classification performance than AMULET, as measured by the F1 score. Furthermore, SEBULA provides reasonably accurate estimates of multiplet proportions across all datasets (Fig 3C), suggesting that the probabilities derived from SEBULA's semi-parametric model exhibit good calibration properties. These multiplet proportion estimates can also serve as a useful quality control (QC) indicator for snATAC-seq data.

For comparison, we also implemented and evaluated an alternative negative binomial mixture model (NBMM), in which the HCLC distribution of singlets is explicitly modeled by a negative binomial distribution (Section 1 of S1 Text). However, when applied to the simulated artificial doublet datasets, the NBMM produced poorly calibrated classification probabilities: both the estimated singlet proportions and individual posterior probabilities for the vast majority of cells were heavily concentrated near 1 (i.e., indicating singlets with near certainty) across all simulation settings (S2 Fig). We interpret this as evidence that the negative binomial assumption is inadequate for modeling singlet distributions in real data. While the negative binomial distribution addresses some limitations of the Poisson model used by AMULET (such as the inability to accommodate overdispersion or zero inflation), it appears overly permissive for large HCLC values and thus fails to effectively separate multiplets from singlets. From another perspective, this experiment further highlights SEBULA's key advantage in accurately modeling the singlet and multiplet distributions in real data.

## Multiplet detection in trimodal single-cell data

We applied SEBULA to seven trimodal single-cell datasets generated by DOGMA-seq experiments and benchmarked its performance against state-of-the-art multiplet detection methods. DOGMA-seq simultaneously profiles the transcriptome (scRNA-seq), chromatin accessibility (scATAC-seq), and cell surface protein expression (via antibody-derived tags, or ADTs) in single cells. We collected seven benchmarking datasets generated from human peripheral blood T cell experiments, including one dataset from Xu *et al.* [16] (GSE200417) and six from Hu *et al.* [14]. In all cases, high-confidence binary labels for multiplet classification were obtained using cell hashing [7] and were treated as ground truth for evaluation.

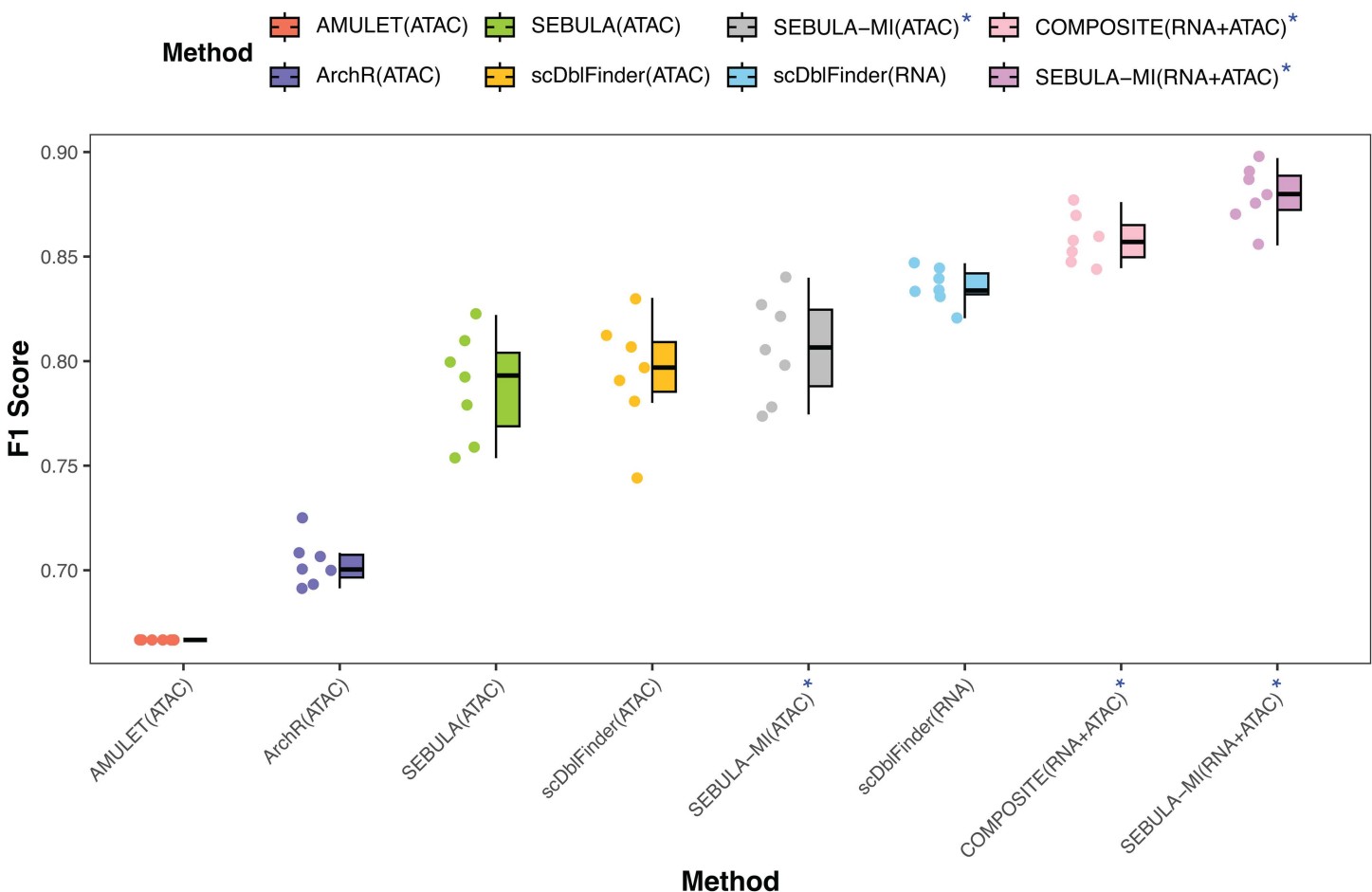

**Fig 4. Benchmarking SEBULA against existing multiplet detection methods using seven DOGMA-seq datasets.** Boxplots show F1 scores for each multiplet detection method evaluated across seven DOGMA-seq datasets. Multiplet labels in all datasets are derived from cell hashing. SEBULA-MI denotes SEBULA's results from multiomic integration. AMULET(ATAC), ArchR(ATAC), SEBULA(ATAC), and scDblFinder(ATAC) denote methods applied using only ATAC-seq data, whereas scDblFinder(RNA) denotes scDblFinder applied using only RNA-seq data. SEBULA-MI(ATAC)* denotes SEBULA integrating its ATAC-seq analysis with evidence from scDblFinder's ATAC-seq analysis. COMPOSITE(RNA+ATAC)* denotes COMPOSITE applied to both RNA-seq and ATAC-seq data. SEBULA(RNA+ATAC)* denotes SEBULA integrating its ATAC-seq analysis with evidence from scDblFinder's RNA-seq analysis. Each point represents the result for a single dataset; boxes span the interquartile range with the median marked by a horizontal line, and whiskers extend to 1.5× the interquartile range.

Among the methods that rely solely on the scATAC-seq modality, SEBULA consistently outperformed AMULET and ArchR in terms of F1 score and performed comparably to scDblFinder (Fig 4). Notably, both ArchR and scDblFinder classify multiplets using *in silico* doublet generation and leverage multiple features from scATAC-seq data, excluding HCLC. The performance gap between SEBULA and AMULET corroborates the findings from our synthetic data simulations. We further hypothesize that the benchmarking datasets are enriched for homotypic multiplets, a setting in which the HCLC statistic is particularly informative. This likely explains SEBULA's advantage over ArchR and, in some cases, over scDblFinder.

Next, we applied the scheme in Eqn. (3) to integrate the classification probabilities derived from scDblFinder and SEBULA within the ATAC-seq modality, effectively combining the evidence from ATAC-seq features leveraged by scDblFinder with that from the HCLC statistic. This evidence ensemble achieved the best overall performance in the ATAC-seq modality across all benchmarking datasets (Fig 4).

We subsequently extended the evidence integration across data modalities by combining scRNA-seq-based inference from scDblFinder with ATAC-seq-based inference from SEBULA, again using Eqn. (3) to compute per-cell classification probabilities. Notably, scRNA-seq-based inference alone achieved higher sensitivity and specificity for multiplet detection than scATAC-seq-based inference. Nevertheless, integrating evidence from both modalities led to substantial improvements over either modality alone, underscoring the complementary value of multimodal data. Remarkably, the ensemble classifier's performance was on par with (and in some cases slightly better than) COMPOSITE, a state-of-the-art method specifically designed for cross-modality multiplet detection.

Finally, we evaluated the computational efficiency of SEBULA relative to ArchR and scDblFinder (ATAC) across representative benchmark datasets (Section 4 of S1 Text). SEBULA required substantially shorter runtime and lower memory usage for multiplet detection (S3 Fig). Given the HCLC data, mixture model fitting and posterior computation in SEBULA are computationally lightweight. These results demonstrate that SEBULA provides an efficient alternative to simulation-based approaches for large-scale single-cell datasets.

## Discussion

SEBULA implements an efficient and robust computational model for detecting multiplets using fragment-level information in scATAC-seq data. It demonstrates improved sensitivity and specificity compared with competing approaches across extensive simulations and real-data analyses. SEBULA's probabilistic output can be seamlessly integrated with other evidence from diverse features and modalities, enabling even more effective multiplet detection.

For snATAC-seq data, SEBULA focuses on a single feature, HCLC, which we demonstrate to be highly informative for distinguishing multiplets from singlets. Our main contribution in this context is the semi-parametric modeling of the singlet distribution within an empirical Bayes framework. Because of the complex nature of snATAC-seq experiments and the construction of the HCLC statistic (which involves explicit censoring of genomic loci), we find that standard parametric distribution families cannot adequately fit the empirical singlet data. The semiparametric strategy in SEBULA assumes that low HCLC values are dominated by singlets. By applying the Box-Cox transformation, which effectively normalizes HCLC count data, we expect that a normal density can adequately characterize the singlet distribution in the low HCLC range due to the central limit theorem. This strategy proves effective in both our simulations and real data applications. Compared with alternative nonparametric approaches such as artificial doublet simulation (which learn the empirical multiplet distribution from synthetic hybrids constructed from observed cells), the assumption that SEBULA relies on is weaker and more practical. As a result, we observe that SEBULA maintains strong performance even when multiplets become abundant or dominant (S6 Fig). Under similar conditions, distributions learned via artificial doublet simulations can become substantially skewed, thereby affecting their ability to detect multiplets.

SEBULA uses a naïve Bayes framework to integrate evidence across different data features and modalities. This framework encourages a divide-and-conquer strategy for accommodating increasingly complex genomic data structures: Rather than attempting to jointly model all features across modalities simultaneously, it can be more cost-effective to design analysis modules targeting individual features and subsequently combine the evidence from these modules. SEBULA can operate on evidence represented in various forms, including posterior probabilities, likelihood functions/ Bayes factors, $z$-scores, and $p$-values. As a naïve Bayes classifier, SEBULA adopts the naïve independence assumption, i.e., that features are independent conditional on the singlet/multiplet labels. We acknowledge that this assumption may be overly strong and unrealistic for real data. To assess its validity in real data, we examined the empirical correlations among the integrated features across RNA and ATAC modalities in the DOGMA-seq datasets and found that the naïve independence assumption is not severely violated (Section 5 of S1 Text and S7 Fig). Furthermore, it has been widely recognized that even when the naïve independence assumption is not strictly satisfied, the decision boundaries for classification produced by the naïve Bayes classifier can still be optimal [25–27]. The classification performance by SEBULA from the real data analysis appear consistent with this understanding.

 

Finally, we acknowledge the limitations of the current SEBULA implementation and are committed to further improvements in future work. First, although SEBULA is efficient given the HCLC statistics, extracting HCLC statistics from ATAC-seq fragment files can be computationally expensive. Developing more efficient algorithms that optimize both CPU and memory usage for this preprocessing step could substantially accelerate the overall SEBULA workflow. Second, as our primary focus in this work is evaluating SEBULA's classification performance, further attention should be devoted to the calibration of the classification probabilities (i.e., whether the reported probabilities align with the observed frequencies of outcomes). This is important because well-calibrated probabilities can facilitate more precise control of data quality in single-cell experiments. While the current implementation exhibits certain properties consistent with well-calibrated probabilities, for example, the estimated multiplet proportions in our simulated datasets are generally accurate (S2 Fig), a more detailed investigation and methodological development are needed to ensure stronger calibration guarantees.

## Methods

### Pre-processing and HCLC generation

To generate the high-coverage locus count (HCLC) statistics required for SEBULA, we largely follow the preprocessing procedures established in AMULET, which consist of two main stages.

In the first stage, we process fragment files containing all read alignments along with a set of confidently called cell barcodes. For each autosomal chromosome in each cell, we retain fragments with insert sizes less than 900 base pairs. We then identify and record high-coverage loci (defined as genomic intervals overlapped by more than two fragments within a single cell) using a sweep-line algorithm.

In the second stage, we aggregate the high-coverage loci identified across all cells to create a non-redundant set of genomic intervals for each chromosome. This is achieved using a greedy interval-merging algorithm. Based on these merged loci, we construct a binary matrix in which each row corresponds to a high-coverage locus and each column corresponds to a cell. An entry in the matrix is set to 1 if the cell contains a fragment overlapping the corresponding locus, and 0 otherwise. The HCLC for each cell is then computed as the sum of entries in the corresponding column.

### Semi-parametric modeling of HCLC

Let $X = x_1, \ldots, x_N$ denote the HCLC values for $N$ cell barcodes in an experiment. We first exclude cells with HCLC strictly below a predefined threshold $x_{\min}$, assuming that such cells are predominantly singlets. In the Method part, we provide a computational procedure for selecting an optimal value of $x_{\min}$ based on the overall goodness-of-fit of the SEBULA model. However, we find that SEBULA's results are generally robust when $x_{\min} \in [0, Q_1]$, where $Q_1$ denotes the 25th percentile. We denote the set of filtered HCLC values by $X_T$ and the number of retained cells by $N_T$. For each $x_j \in X_T$, we apply a variance-stabilizing and Box-Cox transformation [28],

$$z_j = \frac{x_j^\lambda - 1}{\lambda},$$

where the parameter $\lambda$ is determined by maximizing the Box-Cox log-likelihood. SEBULA models the Box-Cox transformed HCLC data, $Z_T$, using a two-component mixture distribution:

$$f(z) = \pi_0 f_0(z) + \pi_1 f_1(z),$$

where $f_0$ is assumed to be a normal density with (unknown) mean, $\delta_0$, and variance, $\sigma_0^2$.

**Mixture density estimation via poisson spline smoothing.** We estimate the mixture density $f(z)$ using a smoothed histogram of the truncated variable $Z_T$, partitioned into $K$ equally spaced bins (default $K = 120$). For $i = 1, \ldots, K$, let $y_i$ denote the observed count in the $i$th bin and $c_i$ the center of the bin.

Following Efron [23,29,30], we treat $y_i$ as independent Poisson observations and fit a generalized linear model (GLM) with natural spline basis of degree 7. Specifically, we assume

$$\log f(z) = \beta_0 + \sum_{k=1}^{7} \beta_k \, ns_k(z),$$

where $ns_k(z)$ denotes the $k$th basis function of the natural spline. This yields a smooth estimate $\hat{f}(z)$ of the mixture density. The fitted values at bin centers, $\hat{f}_i = \hat{f}(c_i)$, are normalized to produce a discrete approximation of the density:

$$\hat{\rho}_i = \frac{\hat{f}_i}{N_T}.$$

A smooth estimate of the cumulative distribution function (CDF), $\hat{F}(z)$, is subsequently obtained by linearly interpolating the bin centers $c_i$ against the cumulative sums of $\hat{\rho}_i$.

**Singlet density estimation by central matching.** We apply Efron's central matching method [23] to estimate the singlet density $f_0(z)$ and the singlet proportion $\pi_0$ in the transformed HCLC data. The key assumption here is that the vast majority of the transformed HCLC values within a predefined central window correspond to singlet cells. (By default, we specify the central window spanning the 20th to 60th percentiles of the empirical distribution of $Z_T$.) By isolating data within this range, we fit a quadratic model to the log of the sub-density $f_0^+(z) := \pi_0 f_0(z)$ as follows:

$$\log\{\pi_0 f_0(z)\} = \beta_0 + \beta_1(z - c_{max}) + \beta_2(z - c_{max})^2, \tag{4}$$

where $c_{max}$ is the center of the histogram bin where the estimated mixture density $\hat{f}(z)$ attains its maximum.

Assuming $f_0(z) \sim \mathcal{N}(\delta_0, \sigma_0^2)$, the log sub-density expands as

$$\log\{\pi_0 f_0(z)\} = \log(\pi_0) - \frac{1}{2}\left(\frac{\delta_0^2}{\sigma_0^2} + \log(2\pi\sigma_0^2)\right) + \frac{\delta_0}{\sigma_0^2}z - \frac{1}{2\sigma_0^2}z^2, \tag{5}$$

The least squares estimates, $\hat{\beta}0, \hat{\beta}1, \hat{\beta}2$, from the fit of (4) induce the estimates of $\hat{\delta}0$ and $\hat{\sigma}0$ for the normal density of the singlet distribution:

$$\hat{\delta}_0 = c_{max} - \frac{\hat{\beta}_1}{2\hat{\beta}_2}, \qquad \hat{\sigma}_0 = \left(-2\hat{\beta}_2\right)^{-1/2}, \quad \hat{\beta}_2 < 0.$$

An unnormalized estimate of the singlet count in each histogram bin $i$ is then computed by

$$\tilde{f}_i = \exp\left\{\hat{\beta}_0 + \hat{\beta}_1(c_i - c_{max}) + \hat{\beta}_2(c_i - c_{max})^2\right\}, \qquad i = 1, \ldots, K,$$

where $\hat{f}_i$ denotes the fitted count from the Poisson spline model at the $i$th bin center, $c_i$.

Finally, the singlet proportion $\pi_0$ is estimated by normalizing the total fitted singlet density:

$$\hat{\pi}_0 = \frac{\sum_{i=1}^{K} \tilde{f}_i}{\sum_{i=1}^{K} \hat{f}_i}.$$

Note that $\hat{\pi}_0$ represents the proportion of singlets among the cells after initial filtering. To report the estimate of the singlet proportion in the *entire* original library, we use

$$\tilde{\pi}_0 = 1 - \frac{N_T}{N}(1 - \hat{\pi}_0).$$

**Optimizing truncation point $x_{\min}$.** In practice, we find that filtering out zero and low HCLC values can substantially improve the efficiency of mixture distribution estimation. Although the final results for multiplet classification are generally robust to the specific choice of the threshold $x_{\min}$, provided that $x_{\min} \geq 0$, we propose a simple computational procedure to aid in selecting an $x_{\min}$ value to optimize the performance of SEBULA.

The proposed procedure evaluates the goodness-of-fit between the transformed HCLC values within a specific range and the estimated normal distribution using cross-entropy loss. Specifically, after filtering the data and fitting the mixture model, we define

$$S = \{z_j \in Z_T : \hat{\delta}_0 - \hat{\sigma}_0 \leq z_j \leq \hat{\delta}_0 + \hat{\sigma}_0\},$$

and compute the cross-entropy for the corresponding $x_{\min}$ as

$$CE(x_{\min}) = -\frac{1}{|S|}\sum_{z_j \in S}\log \hat{f}_0(z_j).$$

The cross-entropy measures the discrepancy between the fitted normal distribution and the empirical distribution of the data; lower values indicate a better fit. Therefore, selecting an $x_{\min}$ that minimizes the cross-entropy supports the validity of the central matching assumption.

Because the mixture fitting procedure is computationally efficient, SEBULA can quickly evaluate cross-entropy values over a range of $x_{\min}$ settings. To balance improved goodness-of-fit with retaining sufficient data for stable estimation of the mixture model, SEBULA constructs a trace plot of $CE(x_{\min})$ and selects the elbow point as the optimal threshold (S4 Fig).

To further assess SEBULA's robustness to the choice of truncation threshold, we conducted a sensitivity analysis by varying $x_{\min}$ over a broad range. Classification performance metrics, including F1 score, AUROC, and AUPRC, as well as the called multiplet rate, remained stable over practical threshold choices (S5 Fig). These results indicate that SEBULA is not highly sensitive to moderate variations in $x_{\min}$ and that the proposed cross-entropy-based selection procedure provides a reasonable and stable operating point.

## Multiplet classification

Based on the overall procedure, we compute the posterior probability that cell $j$ is a multiplet as

$$p_j := P(\text{ cell } j \text{ is a multiplet} \mid X) = \begin{cases} 0, & \text{if } x_j \leq x_{\min}. \\ 1 - \frac{\hat{\pi}_0 \hat{f}_0(z_j)}{\hat{f}(z_j)}, & \text{otherwise.} \end{cases}$$

Thresholding the $p_j$ values is a standard classification approach in machine learning practice. By default, we classify cells with $p_j > 0.8$ as multiplets, corresponding to a local false discovery rate threshold of 20%.

Alternatively, we also implement a classification procedure that controls the overall false discovery rate (FDR), primarily for comparison with AMULET. Note that the estimated densities $\hat{f}_0$ and $\hat{f}$ induce corresponding cumulative distribution functions $\hat{F}_0$ and $\hat{F}$. For a decision rule that classifies all cells with Box-Cox transformed values $z \geq t$ as multiplets, the estimated FDR at threshold $t$ is given by

$$\widehat{\mathrm{Fdr}}(t) = \hat{\pi}_0 \frac{1 - \hat{F}_0(t)}{1 - \hat{F}(t)}.$$

This approach allows users to pre-specify a target FDR level (e.g., 20%) and then identify the corresponding $z$ cutoff directly from the data.

**Probabilistic evidence integration**

To integrate probabilistic evidence from HCLC with evidence derived from other data features and/or modalities, we adopt a general Bayesian framework for sequentially updating posterior multiplet probabilities.

We regard the existing posterior probability for a target cell, $P(\mathrm{multiplet} \mid \boldsymbol{y})$, as the new prior for re-analyzing the HCLC data. We then recover the HCLC likelihood for each cell from the earlier HCLC-only analysis. Specifically, we compute the marginal likelihood of the HCLC data, i.e., the Bayes factor, for cell $j$ as

$$\mathrm{BF}_j(x) = \frac{p_j}{1 - p_j} \Big/ \frac{\tilde{\pi}_0}{1 - \tilde{\pi}_0},$$

where $p_j$ is the posterior multiplet probability based on HCLC alone, and $\tilde{\pi}_0$ is the estimated proportion of singlets in the original library.

Assuming conditional independence between $\boldsymbol{x}$ and $\boldsymbol{y}$, i.e.,

$$P(\boldsymbol{x} \mid \mathrm{multiplet}) = P(\boldsymbol{x} \mid \mathrm{multiplet}, y), \tag{6}$$

The updated posterior probability incorporating HCLC evidence is given by the Bayes rule:

$$P(\mathrm{multiplet} \mid \boldsymbol{x}, \boldsymbol{y}) \propto P(\mathrm{multiplet} \mid \boldsymbol{y}) \cdot \mathrm{BF}_j(\boldsymbol{x}). \tag{7}$$

Note that the conditional independence assumption in (6) is the naïve independence assumption used in the naïve Bayes classifier. While this assumption may not hold exactly in practice, empirical evidence suggests that it is approximately valid (Section 5 of S1 Text), particularly when integrating data across distinct modalities. Additionally, the same naïve Bayes procedure can be applied to integrate evidence in the form of $p$-values or $z$-scores (see Section 3 of S1 Text).

## Supporting information

**S1 Text. Supplementary methods and results.**
(PDF)

**S1 Fig. Performance comparison across simulation scenarios with varying doublet proportions.** (A) Area under the receiver operating characteristic curve (AUROC). (B) Area under the precision-recall curve (AUPRC) for SEBULA (red) and AMULET (blue) evaluated on synthetic datasets with known doublet labels. Each boxplot summarizes results across 20 replicates at a given doublet proportion (ranging from 5% to 25%).
(EPS)

**S2 Fig. Performance comparison of negative bionomial model and SEBULA using simulated doublet datasets.** (A) Boxplots of F1 score, AUROC, and AUPRC across simulated datasets with varying singlet proportions (0.75–0.95), comparing AMULET (blue), SEBULA (red), and a two-component negative binomial (NB) mixture model fitted via EM (orange). SEBULA consistently outperforms AMULET across all scenarios and achieves comparable

ranking metrics to the NB mixture model. (B) Estimated singlet proportions ($\hat{\pi}_0$) from the NB mixture model across simulation settings. Despite varying ground-truth multiplet frequencies, the NB model consistently estimates $\hat{\pi}_0$ values near one, indicating poor calibration. (C) Histogram of posterior singlet probabilities from the NB mixture model under a representative simulation setting. Predicted probabilities cluster tightly near 1, offering limited resolution for distinguishing doublets. These results highlight the limitations of fully parametric models in capturing classification uncertainty when assumptions are violated and support the use of SEBULA's semi-parametric framework for robust doublet detection.

(EPS)

**S3 Fig. Comparison of computational efficiency of methods across representative datasets.** Runtime (seconds) and peak memory usage (GB) of SEBULA, ArchR, and scDblFinder (ATAC) were evaluated on three DOGMA-seq datasets (PB-1, PB-3, and PB-8). SEBULA consistently required lower memory and shorter runtime during the detection stage compared to simulation-based approaches.

(EPS)

**S4 Fig. Selection of truncation point and assessment of null fit in Simulation 20 (25% doublets).** (A) Trace plot showing the cross-entropy (negative average log-likelihood) evaluated within the local window ($\hat{\mu}_0 \pm \hat{\sigma}$) across candidate truncation points. Numbers above each point indicate the number of cells retained after truncation. (B) Distribution of Box–Cox transformed accessibility scores after applying a selected truncation point. The histogram represents the empirical distribution of transformed scores, with overlaid estimated null (singlet) density (blue solid line) and overall mixture density (red dashed line). Vertical dashed lines indicate the estimated null mean ($\hat{\mu}_0$, orange), the central window used for null estimation (green), and the evaluation window ($\hat{\mu}_0 \pm \hat{\sigma}$) used for cross-entropy calculation (purple).

(EPS)

**S5 Fig. Sensitivity of SEBULA to the truncation threshold $x_{\min}$ in the PB-3 dataset.** Performance metrics of SEBULA were evaluated across a range of truncation thresholds $x_{\min}$, which define the minimum high-coverage locus count (HCLC) retained for mixture modeling. Shown are the F1 score, area under the receiver operating characteristic curve (AUROC), area under the precision–recall curve (AUPRC), and the fraction of cells classified as multiplets as functions of $x_{\min}$ (0–11). Across the examined range, F1, AUROC, and AUPRC remain stable with only minor fluctuations, indicating that classification performance is largely insensitive to moderate variations in the truncation threshold. The called doublet rate varies modestly but remains within a narrow range, suggesting stable prevalence estimation.

(EPS)

**S6 Fig. Stress-test simulation under an extreme multiplet loading scenario.** Histogram of Box–Cox–transformed high-coverage locus counts (HCLC) generated from a simulated dataset with an approximately 40% multiplet rate. Singlet cells are shown in pink and simulated doublets in blue. Despite the elevated doublet proportion, singlets continue to form the dominant central peak of the distribution, while doublets are shifted toward larger transformed HCLC values. The red dashed lines indicate the central window used for empirical null estimation in SEBULA. Notably, this region remains largely populated by singlets even under this extreme loading condition, visually supporting the validity of the zero-assumption underlying the central matching procedure.

(EPS)

**S7 Fig. Assessment of dependence between ATAC- and RNA-derived multiplet scores.** Scatter plot of SEBULA ATAC-based scores and RNA-based scores (from scDblFinder) for cells in the PB-3 dataset. Each point represents a single cell.

(EPS)

## Author contributions

**Conceptualization:** Yuntian Wu, Johann E. Gudjonsson, Lam C. Tsoi, Xiaoquan Wen.

**Data curation:** Haoran Hu, Wei Chen.

**Formal analysis:** Yuntian Wu, Xiaoquan Wen.

**Funding acquisition:** Johann E. Gudjonsson, Lam C. Tsoi, Xiaoquan Wen.

**Investigation:** Yuntian Wu, Haoran Hu, Wei Chen, Johann E. Gudjonsson, Lam C. Tsoi, Xiaoquan Wen.

**Methodology:** Yuntian Wu, Lam C. Tsoi, Xiaoquan Wen.

**Software:** Yuntian Wu.

**Writing – original draft:** Yuntian Wu, Haoran Hu, Wei Chen, Johann E. Gudjonsson, Lam C. Tsoi, Xiaoquan Wen.

**Writing – review & editing:** Yuntian Wu, Haoran Hu, Wei Chen, Johann E. Gudjonsson, Lam C. Tsoi, Xiaoquan Wen.

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
