## [Decision Letter · Decision Letter 0]

29 Jan 2026

PCOMPBIOL-D-25-02188

Semi-parametric Empirical Bayes Method for Multiplet Detection in snATAC-seq with Probabilistic Multi-omic Integration

PLOS Computational Biology

Dear Dr. Wen,

Thank you for submitting your manuscript to PLOS Computational Biology. After careful consideration, we feel that it has merit but does not fully meet PLOS Computational Biology's publication criteria as it currently stands. Therefore, we invite you to submit a revised version of the manuscript that addresses the points raised during the review process.

We look forward to receiving your revised manuscript.

Kind regards,

Zhixiang Lin

Academic Editor

PLOS Computational Biology

Shaun Mahony

Section Editor

PLOS Computational Biology

**Additional Editor Comments:**

Please refer to the comments raised by the reviewers.

**Journal Requirements:**

Potential Copyright Issues:

- Figure 1. Please confirm whether you drew the images / clip-art within the figure panels by hand. If you did not draw the images, please provide (a) a link to the source of the images or icons and their license / terms of use; or (b) written permission from the copyright holder to publish the images or icons under our CC BY 4.0 license. Alternatively, you may replace the images with open source alternatives. See these open source resources you may use to replace images / clip-art:

State what role the funders took in the study. If the funders had no role in your study, please state: "The funders had no role in study design, data collection and analysis, decision to publish, or preparation of the manuscript.".

**Reviewers' comments:**

Reviewer's Responses to Questions

**Comments to the Authors:**

Reviewer #1: 1. It would be beneficial to explore and clearly articulate the limitations of the methods used in the study.

2. Consider including additional methodologies to enable a more comprehensive comparison and evaluation.

3. A careful review of the manuscript for minor grammatical errors is recommended to ensure clarity and professionalism.

4. Improving the quality and resolution of the figures would significantly enhance the visual appeal and readability of the paper.

5. The manuscript would benefit from a more cohesive narrative structure. The abstract, introduction, and related work sections should clearly highlight the current challenges and gaps in the literature that the study aims to address.

Reviewer #2: Review on Semi-parametric Empirical Bayes Method for Multiplet Detection in snATAC-seq with Probabilistic Multi-omic Integration

The manuscript presents SEBULA, a statistical framework designed to identify multiplets (droplets containing two or more cells) in single-nucleus ATAC-seq data. The authors identify a critical limitation in existing methods like AMULET, which rely on rigid parametric assumptions (such as Poisson distributions) that fail to capture the overdispersion and multimodality often found in real chromatin accessibility data. To address this, SEBULA utilises a semi-parametric empirical Bayes framework to estimate the singlet distribution directly from the data, specifically leveraging the "high-coverage locus count" (HCLC) metric. Furthermore, the authors introduce a probabilistic integration strategy that combines SEBULA’s output with evidence from other modalities, such as scRNA-seq, via a Bayesian update rule. Benchmarking is performed on simulated data and seven annotated DOGMA-seq datasets, demonstrating that SEBULA generally outperforms AMULET and ArchR, particularly in datasets with higher doublet proportions.

This manuscript addresses a significant quality control challenge in single-cell genomics. The move toward a semi-parametric approach is well-motivated by the specific statistical properties of snATAC-seq data (sparsity and overdispersion). The "divide-and-conquer" modularity of the method, which allows for the probabilistic integration of multiomic data, is a valuable contribution to current multimodal workflows.

Overall, the manuscript addresses a relevant challenge and introduces a method with practical utility. The semi-parametric approach is conceptually sound and empirically strong, and the focus on calibration and false discovery rate control is welcome. The paper is generally well written. Several aspects, however, would benefit from clarification, more systematic evaluation, or fuller justification before publication.

Major comments

The manuscript emphasizes the importance of calibrated probabilities over simple ranking. While the estimation of multiplet prevalence (pi_0) is presented, the paper would greatly benefit from a more detailed analysis of the calibration itself. Reliability diagrams (calibration plots) comparing predicted probabilities to observed multiplet frequencies in the ground-truth datasets or quantitative scores such as expected calibration error could further strengthen the claim that SEBULA yields "well-calibrated posterior probabilities".

The method relies on the "zero assumption," specifically that the central mode of the data is dominated by singlets and that singlets are substantially more abundant than multiplets. While this holds for standard experiments, "super-loaded" runs can have extremely high doublet rates. The authors could discuss or simulate the performance of SEBULA in scenarios with very high doublet rates to define the breaking point of this assumption. For example, demonstrations of failure modes and guidance for identifying them would strengthen reader confidence.

The current validation focuses heavily on benchmarking metrics (F1 scores, ROC curves). To increase the relevance of the manuscript, it would be beneficial to demonstrate the impact of using SEBULA on a downstream biological application or something similar. Additionally, the authors could explicitly discuss the limitation that the method requires fragment files and cannot run on processed count matrices alone, which may limit its applicability for some users.

The multimodal integration uses a naïve Bayes assumption. Although the authors note the limitations, the performance comparisons against COMPOSITE may not be fully fair unless the independence assumption is examined. An analysis when naïve Bayes fusion may inflate or deflate confidence due to dependence would be valuable.

Figure 1 is intended to provide an overview of the framework, but it currently lacks context regarding the problem setup. The figure should ideally illustrate the data acquisition or the biological origin of multiplets to set the stage. A "zoom-in" approach detailing the different stages (preprocessing vs. detection vs. integration) would make the workflow easier to parse visually.

Can the authors comment on further literature and their relevancy such as OmniDoublet (and the references in there such as Scrublet and DoubletFinder)?

Minor comments

The introduction establishes the problem well, but the focus on the integration of multiple modalities appears somewhat late. Given that SEBULA-MI (the integrated version) yields the best performance, this motivation should be highlighted earlier in the introduction to better frame the paper’s contributions.

Detecting homogeneous doublets is rather difficult in general, what would be the impact on a downstream analysis of missing them?

The Box–Cox transformation is a key component of the pipeline, yet several choices are presented as default without a clear rationale. The transformation may blur biologically meaningful differences across cells with extreme HCLC.

The abbreviation "SEBULA" is introduced without a clear explanation of what the acronym stands for in the main text (it appears to be derived from the title, but this should be explicit).

In Figure 2, it is unclear which panels correspond to labels A and B. Moreover, the text claims the singlet distribution is "multimodal". However, in the density plot (Figure 2A), the singlet distribution appears as a single sharp spike. If multimodality exists, the x-axis scale or visualization needs to be adjusted to make this visible to the reader.

Runtime comparisons against the other methods would highlight SEBULA’s practical usability at scale.

A few minor spelling errors, e.g., “applyed,” (l.172) “effectively” (l.168), should be corrected.

Reviewer #3: This manuscript introduces SEBULA, a novel computational tool for detecting multiplets (doublets) in single-nucleus ATAC-seq (snATAC-seq) data. The authors identify a gap in current count-based methods (e.g., AMULET), which rely on rigid parametric assumptions (such as Poisson distributions) that often fail to capture the true data distribution. SEBULA addresses this by employing a semi-parametric empirical Bayes framework that estimates the "singlet" distribution directly from the data using statistical techniques including Box-Cox transformation, central matching, splines, and the method of moments. Furthermore, SEBULA offers a flexible Bayesian framework to integrate its predictions with other modalities (e.g., scRNA-seq), enabling multi-omic doublet detection. Overall, the manuscript is methodologically sound and clearly written. My detailed comments follow.

Major:

1. The method uses a truncation threshold, x_min. Although an optimization procedure (minimizing cross-entropy) is provided and the Methods section claims robustness, including a sensitivity analysis (e.g., a plot of F1 scores as x_min varies) would better demonstrate SEBULA's robustness.

2. HCLC (High-coverage loci) is defined according to AMULET’s preprocessing. As HCLC is an critically important concept to the SEBULA method, it would be helpful to briefly clarify whether this preprocessing step is computationally expensive or if SEBULA includes an optimized preprocessor, as this step is often the bottleneck in read-based methods.

3. The integration of ATAC and RNA signals relies on a "naïve Bayes" assumption of conditional independence between the modalities. While the authors acknowledge this is an approximation, the manuscript would benefit from a brief analysis or plot (perhaps in Supplementary Materials) showing the correlation between HCLC scores and RNA-based doublet scores within the singlet population. If the correlation is low, this would empirically justify the independence assumption.

Minor

1. Although the authors claim the software is "computationally efficient," the Results section lacks specific data to support this. Including a brief comparison of runtime and memory usage between SEBULA, AMULET, and ArchR on a representative dataset would be beneficial.

2. Figure 1 shows that SEBULA can integrate ATAC and RNA data. However, it is unclear whether SEBULA supports integration of more than two modalities, as newer sequencing technologies like DOGMA-seq and TEA-seq profile ATAC, RNA, and surface protein. If so, the Methods section and Figure 1 should be updated accordingly.

3. Rank-preserving Box-Cox transformation: this transformation always preserves ranks as long as \(\lambda\) is fixed. It is unclear why the term “rank preserving” needs to be emphasized.

4. I noticed several typos:

Page 12, line 172: "applyed" should be "applied".

Page 12, line 174: "profiles" should be "profiling".

Page 12, line 189: "effectivey" should be "effective".

Page 16, line 245: "provieds" should be "provides".

**Have the authors made all data and (if applicable) computational code underlying the findings in their manuscript fully available?**

Reviewer #1: None

Reviewer #2: Yes

Reviewer #3: Yes

PLOS authors have the option to publish the peer review history of their article (what does this mean?). If published, this will include your full peer review and any attached files.

Reviewer #1: No

Reviewer #2: No

Reviewer #3: No

**Figure resubmission:**
---

## [Decision Letter · Decision Letter 1]

9 Apr 2026

Dear Wen,

We are pleased to inform you that your manuscript 'Semi-parametric Empirical Bayes Method for Multiplet Detection in snATAC-seq with Probabilistic Multi-omic Integration' has been provisionally accepted for publication in PLOS Computational Biology.

Best regards,

Zhixiang Lin

Academic Editor

PLOS Computational Biology

Shaun Mahony

Section Editor

PLOS Computational Biology

Reviewer's Responses to Questions

**Comments to the Authors:**

Reviewer #1: The authors addressed all my concerns and the paper can be accepted now

Reviewer #2: I appreciate the effort made by the authors. All my major concerns are adressed.

Some minor comments:

The discussion introduces some new results and refers then to the supplement. These insight should ideally already be incorporated into the results part.

Some minor typos remain

- line 207: "Eqn.(3)" instead of "Eqn. (3)"

- line 231: “Nebula“ instead of “Sebula“

Reviewer #3: Thank you for the revision. The updated manuscript has satisfactorily addressed all of my comments.

**Have the authors made all data and (if applicable) computational code underlying the findings in their manuscript fully available?**

Reviewer #1: None

Reviewer #2: Yes

Reviewer #3: Yes

PLOS authors have the option to publish the peer review history of their article (what does this mean?). If published, this will include your full peer review and any attached files.

Reviewer #1: No

Reviewer #2: No

Reviewer #3: No

---

## [Editor Report · Acceptance letter]

PCOMPBIOL-D-25-02188R1

Semi-parametric Empirical Bayes Method for Multiplet Detection in snATAC-seq with Probabilistic Multi-omic Integration

Dear Dr Wen,

I am pleased to inform you that your manuscript has been formally accepted for publication in PLOS Computational Biology. Your manuscript is now with our production department and you will be notified of the publication date in due course.

With kind regards,

Anita Estes
